# The Rehabilitative Effects of Virtual Reality Games on Balance Performance among Children with Cerebral Palsy: A Meta-Analysis of Randomized Controlled Trials

**DOI:** 10.3390/ijerph16214161

**Published:** 2019-10-28

**Authors:** Jinlong Wu, Paul D. Loprinzi, Zhanbing Ren

**Affiliations:** 1Department of Physical Education, Shenzhen University, Shenzhen 518061, China; 1800371011@email.szu.edu.cn; 2Department of Health, Exercise Science and Recreation Management, University of Mississippi, Oxford, MS 38655, USA; pdloprin@olemiss.edu

**Keywords:** virtual reality games, balance, cerebral palsy

## Abstract

This research aims to evaluate the effect of virtual reality (VR) games on balance recovery of children with cerebral palsy (CP) by quantitatively synthesizing the existing literature, and to further determine the impact of VR game intervention (the duration of each intervention, intervention frequency, intervention cycle, and total intervention time) on the balance recovery of children with CP. To this end, relevant literature up until 3 August 2019 was retrieved from Chinese databases (CNKI and Wanfang Data) and the databases in other languages (Web of Science, Pubmed, EBSCOhost, Informit, Scopus, Science Direct, and ProQuest), and bias analysis was conducted with the PEDro scale in this research. Randomized controlled trials (RCTs) were selected and underwent meta-analysis, and combined effect size was calculated with a random effects model. The results showed that VR games may improve the balance of children with CP (Hedge’s g = 0.29; 95% CI 0.10–0.48), and no significant influence of the intervention on balance of children with CP was shown in the subgroup analysis. In conclusion, VR games played a positive role in the improvement of balance of children with CP, but these results should be viewed with caution owing to current methodological defects (difference in measurement, heterogeneity of control groups, intervention combined with other treatments, etc.).

## 1. Introduction

Cerebral palsy (CP) is a type of neurodevelopmental disorder that appears during infancy or early childhood, caused by brain injury or developmental defects [1]. According to the World Health Organization, the incidence of CP in developed countries ranges from 0.2% to 0.3%, making it a major problem for public health [2]. Damage to the central nervous system may cause secondary injuries to children with CP, including physical spasm, amyotrophy/weak muscle tone, skeletal deformity, myasthenia, and developmental coordination disorder [3,4], which results in balance dysfunction. Consequently, the CP children who have balance impairment are more likely to experience increasing numbers of falls and develop more limited motor skills [5]. Well-established balance ability is a fundamental element helping individuals to learn and acquire a different level of motor skills [6]. Furthermore, it is important for children with CP to live an everyday life and take part in social activities and entertainment [6]. Thus, balance function is deemed as an important indicator for the assessment of motor skills of children with CP [7].

Maintaining one’s balance requires continuous and simultaneous data processing from multiple systems (e.g., vestibular, visual, proprioception, and cognitive reintegration) [8,9]. When these systems have been impacted by a neurological injury, balance and balance-contingent activities are affected [10]. Virtual reality (VR) games are whole-body interactive electronic games that feature a three-dimensional display and body perception operated by body movements [11]. These virtual reality games typically use sensory-motor experiences including sight, sound, and touch to simulate environments and activities. Computer screens with special displays are used to present visual imagery while headphones or speakers deliver sound. These immersive virtual environments can be used in balance training by providing continuous visual information to the user that can be used as sensory feedback by the vestibular system to create successful movements. Additionally, the use of virtual avatars performing the same balance training on screen provides users the ability to mirror and modulate their movements in real-time [10]. For children with CP whose cognitive abilities are still in development, however, VR games can engage them and help them carry out repetitive rehabilitation exercises [12]. By the virtue of real-time feedback on human, vision, and feeling from the virtual environment created by VR games, children with CP can interact with the VR environment, and the duration of activities, the intensity of exercises, and times of repetition can be increased [13,14,15].

Current rehabilitation for children with CP focuses on regular rehabilitation training (mainly Bobath and Rood therapeutic methods). However, compared to traditional treatment options, Virtual reality games are a kind of whole-body interactive video game where people can physically immerse in a non-physical world through a three-dimensional display at home [16]. Such an immersive experience in a safe, enjoyable, and playful environment is associated with less fatigue and more relaxation, which may be beneficial for children, including those with CP [17]. Published studies have evaluated the effect of VR games on the upper limb motor skills of children with CP and demonstrated that VR games can improve the upper limb motor skills in this population [18]. Further, a systematic review showed that VR games can enhance the gross motor skills of children with CP, but, notably, these studies lacked a randomized controlled design. A recent systematic review evaluating the balance ability of children with CP found that VR games can, potentially, improve the postural control and balance ability of children with CP [19]. Nevertheless, the evidence of VR games as a therapeutic intervention in children with CP is insufficient, failing to quantitatively reflect these games’ effectiveness of intervening in the development of balance ability of children with CP. Since there is scant attention paid to the balance of children with CP, the main purpose of this research is to explore the effect of VR games on the enhancement of the balance of children with CP, and to further determine the influence of VR games intervention (session length, intervention frequency, intervention cycle, and total intervention time) over the balance of children with CP, so as to lay an important theoretical groundwork for the effect of therapeutic intervention in the balance development of children with CP.

## 2. Methods

This research follows the instructions of the preferred reporting items for systematic review and meta-analysis (PRISMA) statement [20,21].

### 2.1. Retrieval Strategy

The literature in Chinese databases (CNKI and Wanfang Data) and the databases in foreign languages (Web of Science, PubMed, EBSCOhost, Informit, Scopus, Science Direct, and ProQuest) as of 3 August 2019 was retrieved, with “active video game”, “VR game, “video game, “ children with CP” and “children with cerebral palsy” as Chinese keywords, and “exergame”, “AVG”, “active video game”, “active video gaming”, “Wii”, “Play Station”, “Kinect”, “virtual reality”, “cerebral palsy”, “cerebral palsies”, “Little disease”, “infantile palsies”, “spastic diplegia”, “spastic diplegias”, “spastic diplegic”, “spastic hemiplegia”, “spastic hemiplegias”, “spastic hemiplegic”, “spastic quadriplegia”, “spastic quadriplegias”, “spastic quadriplegic”, “postural control”, “trunk control”, “posture”, and “balance” as English keywords.

### 2.2. Inclusion Criteria

The included studies contain Randomized controlled trials (RCTs), and they were included as per the criteria within the Patient Intervention Control Outcome Setting Time (PICOST) framework [22]. Studies were considered eligible if they met the following inclusion criteria. Firstly, studies with a randomized controlled design needed to be published in a peer-reviewed journal. Secondly, study participants were who were at the age of 14 or below and had been diagnosed with CP. Thirdly, virtual reality games were used as the main intervention program to compare non-active and/or active control condition (s). Fourth, balance performance was measured using a valid instrument (the Movement Assessment Battery for Children, the Pediatric Balance Scale, and the Timed Up and Go test). Lastly, useful data (e.g., sample size, mean, and standard deviation) were clearly reported in the published paper or can be retrieved through contacting corresponding author of primary studies. Two researchers independently screened and determined the eligibility of initially retrieved studies against the aforementioned inclusion criteria [23,24]. Notably, the title/abstract was screened, with eligibility agreement that had reached at least 93.5%.Afterwards, Full-text of the remaining items were assessed according to the inclusion criteria, with agreement of 100%.

### 2.3. Quality Assessment

The Kappa index is often used to analyze the consistency of two people’s (or two test methods’) opinions on the same subject. The standard of general judgment is: Kappa index > 0.75 indicates better consistency, 0.75 ≤ Kappa index ≤ 0.4 indicates good consistency, and Kappa index < 0.4 indicates poor consistency. The risk of bias of the included studies was carried out by two independent researchers as per the PEDro scale [25,26,27]. In case of divergence of opinion between the two assessors, a third assessor was consulted. The following 11 criteria were taken into consideration (one point for each criterion): (a) eligibility criteria were specified; (b) subjects were randomly allocated to groups; (c) allocation was concealed; (d) the groups were similar at baseline regarding the most important outcome indicators; (e) there was blinding of all subjects; (f) there was blinding of all therapists; (g) there was blinding of all assessors; (h) measures of at least one key outcome were obtained from more than 85% of the subjects initially allocated to groups; (i) all subjects for whom outcome measures were available received the treatment or, where this was not the case, data for at least one key outcome was analyzed by “intention to treat”; (j) the results of between-group statistical comparisons were reported for at least one key outcome; (k) the study provided both point measures and measures of variability for at least one key outcome. The highest mark for each study was 11: low risk of bias (≥7 or more); moderate risk of bias (5–6); high risk of bias (≤4 or less).

### 2.4. Data Extraction

In the process of retrieval, two researchers independently extracted both descriptive and quantitative data of each included study, and the data were placed in the pre-designed extraction Excel spreadsheet in this order: first author name(s), the year of publication, country, subjects (number, genders, ages), intervention scheme (location(s), method(s), frequency of VR game intervention, time of the intervention), VR game platform(s), VR game type(s), and outcome indicators. Further, the baselines of eligible RCTs and control groups as well as the means (M) and standard deviations (SD) after interventions were also extracted.

### 2.5. Data Analysis

The software Review Manager 5.3 was used in the analysis of outcome indicators of the 11 included studies. Hedge’s g was used to calculate the effect size of each experiment, and the standardized mean difference (SMD) was selected to calculate the effect scale indicators. The effect size shows the influence of VR games on the balance of children with CP, which was calculated through a random effects model and for which 95% CI was applied [28]. The *I*^2^ statistic was used to perform the heterogeneity test: when *I*^2^ ≤ 50%, it indicates no heterogeneity in the studies; when *I*^2^ > 50%, it suggests heterogeneity in the studies, in which case, sensitivity analysis would be conducted to determine the sources of heterogeneity by deleting one study each time. As for subgroup analysis of moderators, intervention schemes were categorized depending on: Session length (15 to 40 min, ≥40 min), intervention frequency (2 to 5 sessions/weekly, ≥5 sessions/weekly), intervention cycle (3 to 12 weeks, ≥12 weeks) and total intervention time (480 to 1000 min, ≥1000 min). Additionally, the funnel plot and Egger’s test were used to assess publication bias [29].

## 3. Results

### 3.1. Retrieval Results

A total of 473 relevant studies were found through article retrieval, including 65 from PubMed, 66 from EBSCOhost, 101 from Web of Science, 52 from InFormit, 34 from Scopus, 96 from Science Direct, and 38 from ProQuest. As for Chinese databases, 9 relevant studies were found through CNKI and 6 found through Wanfang Data. The overall number was reduced to 367 after repeated studies were deleted through NoteExpress 3.2.0. Considering their titles and abstracts, 15 studies were selected, and given the full text of these 15 studies, 4 were excluded due to their failure to meet the inclusion criteria. As a result, 11 RCTs were included in this research (Figure 1).

### 3.2. Study Characteristics

The characteristics of 11 included studies are listed in Table 1. These studies were published from 2010 to 2019, with 7 in English [14,30,31,32,33,34,35], 3 in Chinese [36,37,38] and 1 in Korean [39]. Despite the fact that all subjects were children with CP, there were differences among them (children with paralytic hemiplegia, children with postoperative cerebral palsy, children with hemiplegic cerebral palsy, children with spastic diplegia, children with spastic hemiplegia, and children with spastic quadriplegia). The sample size of each study ranged from 16 to 48 subjects, with a total number of 313. Most of the studies reported the GMFCS levels, genders, and intervention locations of children with CP, whereas two studies did not report the GMFCS levels [32,34], two did not report the genders of subjects [32,33], and two did not report intervention locations [32,34]. The content of intervention was the comparison between VR game schemes and regular rehabilitation schemes as well as the comparison between VR game plus regular rehabilitation schemes and regular rehabilitation schemes. Firstly, the session length from 15 to 40 min (in one of the studies, the duration of each intervention depended on the duration of time). Secondly, intervention frequency was between twice and seven times, with the total intervention time ranging from 480 to 2400 min. Thirdly, intervention cycles also showed large differences, with the minimum cycle being as short as 4 weeks and the maximum reaching 12 weeks.

There were differences in the intervention platforms. Most interventions chose Nintendo as the platform, and the others included the Q4 situational interactive rehabilitation training system, the Xbox Kinect platform, the Biomaster VR training system, and the Scratch software. Compared with intervention groups, control groups included strength training and neurodevelopmental therapy apart from regular rehabilitation. On top of that, in the 11 studies, 8 had only one outcome indicator of balance, and the other 3 [14,30,34] had two outcome indicators of balance.

### 3.3. Methodological Quality

For the quality assessment of included studies, the agreement among assessors was 94.5%. According to the PEDro scale, the methodological quality of all the experimental studies ranged from 5 to 6. The higher the score, the lower the risk of bias (Table 2). Ten studies were deemed to have a moderate risk of bias. Since allocation concealment was not conducted in all the included studies, the quality of included studies decreased. Blinding was not frequently carried out in the studies (it was only conducted in one study [31]), so the score of blinding was deducted for most of the studies.

### 3.4. The Effect of VR Games on the Balance of Children with Cerebral Palsy

#### 3.4.1. The Effect of VR Games on the Balance of Children with Cerebral Palsy for VR Game Groups and Control Groups

This research includes 11 RCT studies with 14 RCT data points, with the mean effect size being 0.29 when all the studies were included (the random effects model), and the funnel plot did not show significant asymmetry (Figure 2) (Egger’s regression intercept = 0.38, *p* = 0.23). Meta-analysis was performed on all RCT data to test the impact upon balance of children with CP, which was measured by different tools, in VR game groups and control groups. The result showed that VR games had a significant effect on the balance of children with CP (small effect, no heterogeneity, Hedge’s g = 0.29, *p* < 0.05, *I*^2^ = 0%) (Figure 3).

#### 3.4.2. Regulatory Impact Analysis

The results of subgroup analysis of VR game intervention in the balance of children with CP are shown in Table 3. It can be seen that because one study [32] did not report specific intervention times and frequencies that were not included in the subgroup analysis, the Session length (*p* = 0.70), intervention frequency every week (*p* = 0.96), Intervention cycle (*p* = 0.33) and the total time of intervention (*p* = 0.34) showed no significant influence.

## 4. Discussion

The results of the present meta-analysis of RCTs evaluating the impact of VR games upon the balance of children with CP showed that VR games had a positive effect on the balance of children with CP compared with traditional rehabilitation methods (Hedge’s g = 0.29; 95% CI 0.10–0.48, *p* < 0.05). However, The study found the session length, intervention frequency every week, Intervention cycle and the total time of intervention showed no significant influence. The reason may be the insufficient number of existing studies, the small number of children with cerebral palsy participating in the experiment and other reasons. The development of various VR games provides new methods of, and approaches to, the treatment of children with CP for clinical staff, and VR games can keep children engaged and doing repetitive rehabilitation exercises. In contrast, traditional dyskinesia treatments require various games and facilities, such as bowling balls, obstacles, and baskets of different heights, and a relatively large space for treatment. Apart from that, traditional rehabilitation treatments also need (a) seasoned therapist(s) to communicate with and encourage children, so as to ensure smooth treatment by increasing children’s excitement and interest [12,40]. Compared with traditional schemes, VR game interventions can be performed in a smaller room, and the interaction of VR game device provides a challenging, encouraging and safe environment, so children with CP who are often unwilling to receive traditional therapy may choose VR game treatment [12,40].

This research holds that the improvement of visual perception, muscle strength, and motor skills on the hemiplegic side by VR games plays the most important role in the enhancement of the balance of children with CP. For example, Bilde et al. [3] found that muscle strength and visual perceptions of children with CP were significantly improved after a VR game intervention. Snider et al. [41] also showed that VR games could improve the visual perception and movement stability of children with CP. Cho et al. [42] found that VR games increased the lower limb strength of children with CP, thereby raising the postural control of these children, enhancing their body symmetry and balance ability.

Chiu et al. [43] noted through the parents’ reports in their study that children’s arms on the hemiplegic side developed new movements (for instance, their forearms could rotate outward). Surprisingly, Rgen et al. [34] found that compared with the children undergoing traditional rehabilitation treatments, the children with CP playing VR games remarkably managed to jump more times with their leg on the hemiplegic side, especially in ski jumping and racing games. The improvement of motor skills on the hemiplegic side can effectively reduce the compensation, keeping body muscles working at the same time and helping with the maintenance of balance.

The advantage of this research lies in the relatively comprehensive study results retrieved, and only RCTs regarding the balance of children with CP. Further, standardized quality assessment tools were adopted, and the heterogeneity of included studies was evaluated. Moreover, the funnel plot and Egger’s test were used to assess publication bias.

Nevertheless, there are limitations in the evaluated studies mainly due to methodological issues. Firstly, there were only 14 RCT data points in our meta-analysis, so the sample size is relatively small, and the included studies may not be as comprehensive as they could be. More well-designed trials with large sample size are needed in future. Some non-peer reviewed papers were not included in the research, so publication bias may exist. Secondly, included studies concerned different VR game intervention schemes and such differences may induce study heterogeneity. The subjects included in this research also varied, though they were all children with CP; the effect of intervention may be different owing to different GMFCS levels. Thirdly, included studies were limited by insufficient description of blinding and randomized control. Experimental procedures of future studies should be transparently reported. In addition, the most important limitation of this research is that the included studies used VR in combination with other rehabilitation programs, so it may be difficult to conclude that improvement was entirely due to VR game intervention. In addition, as the intervention schemes in included studies varied greatly, it is difficult to give specific suggestions on the intervention time and frequency. These issues should be addressed in future studies.

## 5. Conclusions

In conclusion, existing studies show that VR games play a positive role in the improvement of the balance of children with CP. However, the limitations in the current methodology may affect the results of this meta-analysis. More rigorous and reliable RCTs are needed in the future to more accurately determine the improvement effect of VR games on the balance of children with CP as well as the long-term impact of the games upon the balance development of children with CP.

## Figures and Tables

**Figure 1 ijerph-16-04161-f001:**
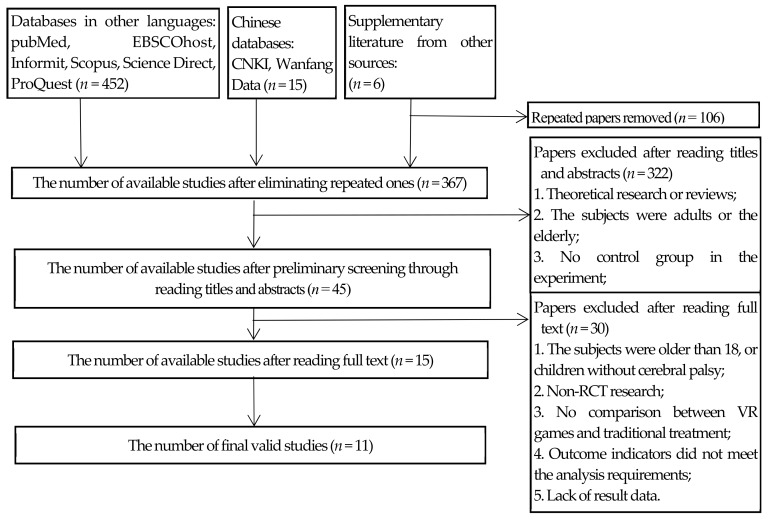
The flow chart of literature screening.

**Figure 2 ijerph-16-04161-f002:**
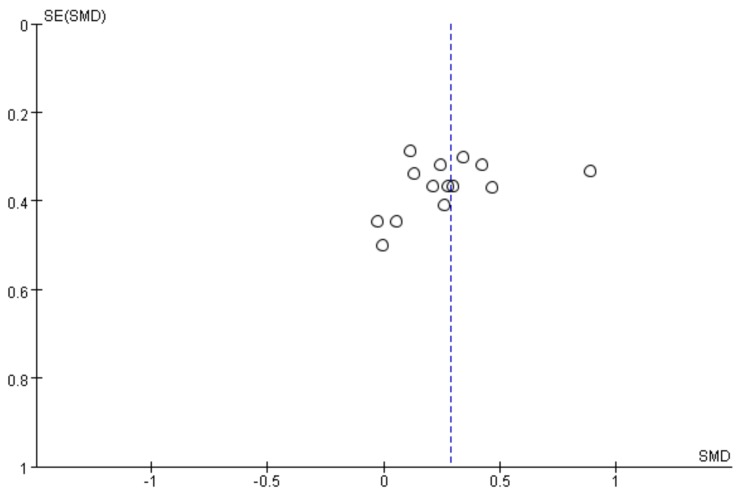
The funnel plot of the effect of VR games on the balance of children with CP.

**Figure 3 ijerph-16-04161-f003:**
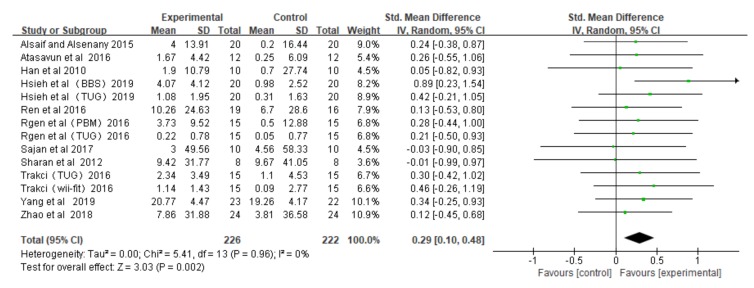
The effect of VR games on the balance of children with cerebral palsy.

**Table 1 ijerph-16-04161-t001:** List of basic characteristics of the included studies.

Researchers	Subjects	Country: Language	GMFCS Level (Number)		Characteristics of Subjects (Number; Genders; Ages)	Intervention Methods; Frequency of the Intervention Group; Total Time of the Intervention Group	Intervention Location	VR Platforms	Types of VR Game	Outcome Indicators
			Experimental Group	Control Group						
Han et al. [39] (2010)	Children with paralytic hemiplegia	South Korea: Korean	L(1):9 L(2):1	L(1):8 L(2):2	20; the experimental group: 10: 5 males (9.50 ± 2.46); the control group: 10: 5 males (8.90 ± 2.37)	Strength training treatment + VRGs treatment; 15 min/time, 3 days/week, 12 weeks in total; 540 min	Hospital	Nintendo wii fit balance board	Warrior posture; wire-waling; rafting in the valley	②
Sharan et al. [32] (2012)	Children with postoperative cerebral palsy	India: English	Not Clear	Not Clear	16; the experimental group: 8 (8.88 ± 3.23); the control group: 8 (10.38 ± 4.41) (NSL)	Depending on game types, 3 days/time, 3 weeks in total	Not clear	Nintendo wii-fit	Depending on children’s abilities	②
Alsaif and Alsenany [33] (2015)	Children with spastic diplegia	Saudi Arabia: English	L(3):20	L(3):20	40; the experimental group: 20; the control group: 20; 6-to-10-year-olds (NSL)	VRGs; 20 min/day, 7 times/week, 12 weeks in total; 1680 min	Home	Nintendo wii fit balance board	Not specified	①
Atasavun et al. [31] (2016)	Children with hemiplegic cerebral palsy	Turkey: English	L(1):9 L(2):3	L(1):10 L(2):2	24; the experimental group: 12: 8 males; the control group: 12: 2 males; 6-to-14-year-olds	Regular rehabilitation treatment + VRGs treatment; 30 min/time, 2 days/week, 12 weeks in total;720 min	Hospital	Nintendo wii-fit	Wii basketball game; Wii football game; Wii boxing game	②
Trakci et al. [14] (2016)	Children with hemiplegic cerebral palsy	Turkey: English	L(1~2):15	L(1~3):15	30; the experimental group: 15: 10 males (10.46 ± 2.69); the control group: 15: 9 males (10.53 ± 2.79)	Neurodevelopmental therapy + VRGs treatment; 20 min/time, twice per week, 12 weeks in total; 480 min	Rehabilitation Centre	Nintendo wii-fit	Slalom; wire-walking; football game	③⑤
Ren et al. [37] (2016)	Children with spastic diplegia	China: Chinese	L(1):8 L(2):11	L(1):7 L(2):9	35; the experimental group: 19: 11 males and 8 females (53.88 ± 13.58); the control group: 16: 9 (56.53 ± 9.67)	Regular rehabilitation treatment + VRGs treatment; 40 min/time,5 times/week, 12 weeks in total; 2400 min	Hospital	Q4 situational interactive rehabilitation training system produced by OPEM	Not specified	④
Rgen et al. [34] (2016)	Children with spastic hemiplegia	Turkey: English	Not Clear	Not Clear	30; the experimental group: 15: 7 males (11.07 ± 2.37); the control group: 15: 7 males (11.33 ± 2.19)	Regular rehabilitation treatment + VRGs treatment; 40 min/time, twice per week, 9 weeks in total; 720 min	Not Clear	Nintendo wii-fit	Ski jumping; snowball fight; jogging; the oblique city; penguin slides; perfect 10; guiseway; header	⑤⑥
Sajan et al. [35] (2017)	Children with double lower limb paralytic and quadriplegic cerebral palsy	India: English	L(1):1 L(2):2 L(3):6 L(4):1	L(2):1 L(3):7 L(4):2	20; the experimental group: 10: 5 males (12.4 ± 4.93); the control group: 10: 6 males (10.6 ± 3.78)	Regular rehabilitation + VRGs treatment; 30 min/time,6 times/week, 3 weeks in total; 540 min	Clinical	Nintendo Wii-fit remote control game	Boxing; tennis	②
Zhao et al. [36] (2018)	Children with hemiplegic, diplegic and quadriplegic cerebral palsy	China: Chinese	L(1):18 L(2):6	L(1):17 L(2):7	48; the experimental group: 24: 11 males (59.38 ± 11.29); the control group: 24: 16 males (54.33 ± 10.93)	Regular rehabilitation treatment + VRGs treatment; 40 min/time, 5 times/week,3 weeks in total; 600 min	Hospital	Active video games on the Xbox Kinect platform	Dance Imitation	②
Hsieh et al. [40] (2018)	Children with diplegic and quadriplegic cerebral palsy	Taiwan, China: English	L(2):10 L(3):6 L(4):4	L(2):10 L(3):5 L(4):5	40; the experimental group: 20: 14 males (7.33 ± 1.31); the control group: 15: 12 males (7.41 ± 1.54)	Regular rehabilitation treatment + VRGs treatment; 40 min/time, 5 times/week,12 weeks in total; 2400 min	Not Clear	computer + joystick (the software Scratch)	Flower watering, monkey eating bananas, killing mosquitoes with a swatter	③⑤
Yang et al. [38] (2019)	Children with spastic diplegia	China: Chinese	L(3):20	L(3):20	45; the experimental group: 23: 9 males (55.53 ± 14.73)l the control group: 22: 12 males (54.85 ± 13.40)	Regular rehabilitation treatment + VRGs treatment; 20 minutes/time, 5 times/week, 12 weeks in total; 1200 min	Hospital	The Biomaster VR training system produced in Guangzhou	Picture matching, skiing, football	③

Note: NSL: The male to female ratio is not mentioned; L:Gross Motor Function Classification System Level of Cerebral Palsy; GMFCS: Gross Motor Function Classification System; VRGs: Virtual reality games; ①: Movement Assessment Battery for Children (M-ABC, the balance test); ②: Pediatric Balance Scale (PBS); ③: Berg Balance Scale (BBS); ④: Nintendo Wii Fit Balance Board Score; ⑤: Timed Up and Go (TUG); ⑥: Pediatric balance measurement (PBM).

**Table 2 ijerph-16-04161-t002:** Methodological quality assessment for the included studies.

Included Studies	A	B	C	D	E	F	G	H	I	J	K	Score
Han et al. [39] (2010)	Yes	Yes	No	Yes	No	No	No	Yes	Not clear	Yes	No	5/11
Sharanl et al. [32] (2012)	Yes	Yes	No	Yes	No	No	No	Yes	Not clear	Yes	No	5/11
Alsaif and Alsenany. [33] (2015)	Yes	Yes	No	Yes	No	No	No	Yes	Not clear	Yes	No	5/11
Atasavun et al. [31] (2016)	Yes	Yes	No	Yes	No	No	Yes	Yes	No	Yes	No	6/11
Trakci et al. [14](2016)	Yes	Yes	No	Yes	No	No	No	Yes	Not clear	Yes	No	5/11
Ren et al. [37] (2016)	Yes	Yes	No	Yes	No	No	No	Yes	No	Yes	No	5/11
Rgen et al. [34](2016)	Yes	Yes	No	Yes	No	No	No	Yes	No	Yes	No	5/11
Sajan et al. [35] (2017)	Yes	Yes	No	Yes	No	No	No	Yes	Yes	Yes	No	5/11
Zhao et al. [36](2018)	Yes	Yes	No	Yes	No	No	No	Yes	No	Yes	No	5/11
Hsieh et al. [30] (2018)	Yes	Yes	No	Yes	No	No	No	Yes	No	Yes	No	5/11
Yang et al. [38](2019)	Yes	Yes	No	Yes	No	No	No	Yes	Not clear	Yes	No	5/11

Note: **A.** eligibility criteria were specified; **B.** subjects were randomly allocated to groups; **C.** allocation was concealed; **D.** the groups were similar at baseline regarding the most important outcome indicators; **E.** there was blinding of all subjects; **F.** there was blinding of all therapists; **G.** there was blinding of all assessors; H. measures of at least one key outcome were obtained from more than 85% of the subjects initially allocated to groups; **I.** all subjects for whom outcome measures were available received the treatment or, where this was not the case, data for at least one key outcome was analyzed by “intention to treat”; **J.** the results of between-group statistical comparisons were reported for at least one key outcome; **K.** the study provided both point measures and measures of variability for at least one key outcome.

**Table 3 ijerph-16-04161-t003:** The regulatory impact analysis of VR game groups and control groups.

Regulatory Impact	Categorical Variables for Sub-Group Analysis	PC	Hedge’s g	95% CI	*I^2^*	Between-Group Difference
						*p*
Session length	15 to 40 min;	7	0.26	−0.01 to 0.53	0%	0.70
	≥40 min	6	0.34	0.07 to 0.60	0%	
Intervention frequency	2 to 5 sessions/weekly;	6	0.27	−0.04 to 0.58	0%	0.96
	≥5 sessions/weekly	7	0.32	0.07 to 0.56	0%	
Intervention cycle	3 to 12 weeks	4	0.15	−0.19 to 0.50	0%	0.33
	≥12 weeks	9	0.36	0.13 to 0.59	0%	
Total time of intervention	480 to 1000 min;	8	0.21	−0.04 to 0.47	0%	0.34
	≥1000 min	5	0.40	0.12 to 0.68	0%	

Note: PC = pair-comparison.

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
