# Peer review of "The Rehabilitative Effects of Virtual Reality Games on Balance Performance among Children with Cerebral Palsy: A Meta-Analysis of Randomized Controlled Trials"

_ijerph, 2019, doi:10.3390/ijerph16214161_

Round 1

Reviewer 1 Report

The study is interesting, as the virtual reality games are more and more often included into standard rehabilitation protocols. One of the patients' groups which are using such additional rehabilitation is cerebral palsy. There are some shortcomings of the study:

definition of CP in the Introduction: the authors stated that CP is caused by a non-progressive brain damage during period from conception till infancy. Just after conception we do not have brains, and the damages which occur later in pregnancy are rather developmental problems. CP is caused by a non-progressive brain damage which occurs around birth time (shortly before, during, and after); in the flowchart (Fig 1) other reasons for exclusion of papers should be somehow explained, as other reasons were clearly described such laconic statement can raise some doubts in the reader (for example maybe such reason for exclusion was lack of positive effect of rehabilitation with VR); table I is difficult to read, I do not have any good suggestion how to improve it, but maybe splitting it into two tables is a solution; the Chinese signs in Table 3 should be removed. This is a paper for international public, and as I understand the inclusion of Chinese key words used for Chinese database search in the Results (and Table 3 is a part of them) they should not be present; in the Discussion and Conclusion sections there is no information that from 347 paper which were extracted during the search only 11 (3 %) matched the criteria. Therefore the conclusions from the study are very limited. This also points to the fact, that the reliable evaluation of VR on the patients' outcome is now difficult, or even impossible.

Author Response

The study is interesting, as the virtual reality games are more and more often included in standard rehabilitation protocols. One of the patients' groups that are using such additional rehabilitation is cerebral palsy. There are some shortcomings of the study: definition of CP in the Introduction: the authors stated that CP is caused by non-progressive brain damage during the period from conception till infancy. Just after conception, we do not have brains, and the damages which occur later in pregnancy are rather developmental problems. CP is caused by non-progressive brain damage which occurs around birth time (shortly before, during, and after);

Response: Thanks for your suggestion. We have revised the definition accordingly, along with relevant references.

in the flowchart (Fig 1) other reasons for exclusion of papers should be somehow explained, as other reasons were clearly described such laconic statement can raise some doubts in the reader (for example maybe such reason for exclusion was lack of positive effect of rehabilitation with VR);

Response: We have revised them as suggested. 

table I is difficult to read, I do not have any good suggestions on how to improve it, but maybe splitting it into two tables is a solution; the Chinese signs in Table 3 should be removed. This is a paper for the international public, and as I understand the inclusion of Chinese keywords used for Chinese database search in the Results (and Table 3 is a part of them) they should not be present;

Response: We have revised the content within Table 1 and removed the Chinese words.

in the Discussion and Conclusion sections, there is no information from 347 paper which was extracted during the search only 11 (3 %) matched the criteria. Therefore the conclusions from the study are very limited. This also points to the fact, that the reliable evaluation of VR on the patients' outcome is now difficult, or even impossible.

Response: We have clearly detailed screening and study selection in the method sections. It should not be considered as limitation.

Reviewer 2 Report

Summary:

A meta-analysis was performed in Chinese and foreign literature to assess virtual reality gamin and how it assisted individuals with cerebral palsy to improve movement coordination and balance.

Major Comments:

Overall, the information the authors present is very good. There are a few items that they need to address in the manuscript.

The title should be “Meta-analysis” not “Mata-analysis”

Table 1 needs reformatting. In its current form, it is not readable.

Minor Comments:

2, lines 7-11: the authors need to provide references for these statements

Author Response

Summary: A meta-analysis was performed in Chinese and foreign literature to assess virtual reality gamin and how it assisted individuals with cerebral palsy to improve movement coordination and balance. Major Comments: Overall, the information the authors present is very good. There are a few items that they need to address in the manuscript. The title should be “Meta-analysis” not “Mata-analysis” Table 1 needs reformatting. In its current form, it is not readable. Minor Comments: 2, lines 7-11: the authors need to provide references for these statements:

Response: Thanks for your suggestion. We have reformat the table in order to be readable, along with corresponding references.

Reviewer 3 Report

This paper presents the results of a Meta-analysis to explore the effect of VR games on the enhancement of the dynamic balance of children with CP, and to further determine the influence of VR game intervention (duration of each intervention, intervention frequency, intervention cycle and total intervention time) over the dynamic balance of children with CP. The idea is laudable, and the results are encouraging, but the authors must rework their article on some points in order to improve it. My comments are listed below.

In the title I think its meta-analysis and not mata-analysis.

Tables 1 and 2 are unpleasant to read. I suggest to the authors to put these tables in landscape format. In addition, I would like the authors to define ‘NSL’ in table 1.

In the section ‘study characteristics’ the authors mentioned GMFCS levels. In my opinion, there is a lot of information missing on this tool. After my research, there are five levels that the authors did not mentioned. I suggest that the authors write a paragraph on this tool and clearly explain the different levels so that the reader can understand more.

In Figure 1 the authors listed as reasons for deleting papers after reading the titles ' other reasons'. I suggest that the authors clarify these reasons.

There is no in-depth discussion on the absence of significant results regarding the comparison between groups (VR game groups and control groups) on regulatory impact (duration of each intervention, intervention frequency, intervention cycle, total time of intervention). I therefore invite the authors to make this discussion.

The authors pointed out several limitations of their work. Overall, these limits considerably reduce the quality of their work. I suggest that they make a short methodological paragraph to overcome these limitations in future studies.

Specify the contribution of each author

Check the names of the authors throughout the text. For example, in the text the reference [30] (page 11 – Line 7) is 'Rgent' while in the bibliographic reference it is'Rgen'. Finally, the bibliographical references do not comply with the standards.

Author Response

This paper presents the results of a Meta-analysis to explore the effect of VR games on the enhancement of the dynamic balance of children with CP, and to further determine the influence of VR game intervention (duration of each intervention, intervention frequency, intervention cycle and total intervention time) over the dynamic balance of children with CP. The idea is laudable, and the results are encouraging, but the authors must rework their article on some points in order to improve it. My comments are listed below. In the title I think its meta-analysis and not mata-analysis.

Response: The title has been revised as suggested. 

Tables 1 and 2 are unpleasant to read. I suggest to the authors to put these tables in landscape format. In addition, I would like the authors to define ‘NSL’ in table 1.

Response: Thanks for your suggestion. We have revised them accordingly. Please read our change in the revised manuscript.

In the section ‘study characteristics’ the authors mentioned GMFCS levels. In my opinion, there is a lot of information missing on this tool. After my research, there are five levels that the authors did not mentioned. I suggest that the authors write a paragraph on this tool and clearly explain the different levels so that the reader can understand more.

Response: As suggested, we have added the number of CP children at different level.

In Figure 1 the authors listed as reasons for deleting papers after reading the titles ' other reasons'. I suggest that the authors clarify these reasons.

Response: We have added the information as suggested in Figure 1.

There is no in-depth discussion on the absence of significant results regarding the comparison between groups (VR game groups and control groups) on regulatory impact (duration of each intervention, intervention frequency, intervention cycle, total time of intervention). I therefore invite the authors to make this discussion.

Response: As suggested, we have added the information.

The authors pointed out several limitations of their work. Overall, these limits considerably reduce the quality of their work. I suggest that they make a short methodological paragraph to overcome these limitations in future studies.

Response: As suggested, we have added more information after reporting the limitation. 

Specify the contribution of each author Check the names of the authors throughout the text. For example, in the text the reference [30] (page 11 – Line 7) is 'Rgent' while in the bibliographic reference it is'Rge.

Response: We have revised it accordingly.

Round 2

Reviewer 3 Report

Good Job